# PureKV: Plug-and-Play KV Cache Optimization with Spatial-Temporal Sparse Attention for Vision-Language Large Models

## Abstract

Vision-Language Large Models (VLLMs) faces significant efficiency challenges when processing high-resolution inputs. The quadratic complexity in attention and autoregressive generation, as well as the constantly growing key value (KV) cache size, severely hinder the prefilling and decoding stages. Recent efforts have attempted to compress KV cache by identifying and pruning KV cache of less important tokens, but these methods typically rely on attention scores to estimate token importance, making them incompatible with efficient attention mechanisms such as FlashAttention and Sparse Attention, which do not explicitly compute attention matrices. Moreover, existing methods overlook how sparse attention, while accelerating the prefilling stage, alters the information structure of the KV cache—thereby compromising the effectiveness of downstream KV cache compression strategies. To address this issue, we propose PureKV, a plug-and-play framework for joint optimization of sparse attention and KV cache compression. We first introduce a KV cache compression strategy that is fully compatible with efficient attention accelerators. Our method utilizes lower layer attention scores to estimate the importance of high layers' KV cache, enabling active pruning without compromising accuracy. In addition, we have designed a Spatial-Temporal Sparse Attention (ST-SpAttn) module specifically tailored for video KV cache compression algorithms. This module combines spatial and temporal attention sparsity to improve the compression efficiency of KV cache optimization algorithms by purifying spatial noise and temporal redundancy in KV cache. At the same time, ST-SpAttn also accelerated the prefilling stage of VLLMs. Extensive experiments on VLLMs (VideoLLaMA2, Qwen2.5-VL) have shown that PureKV achieves 5.0 × KV cache compression and 3.16 × prefill acceleration, with negligible quality degradation. By seamlessly integrating with sparse attention optimization, our work unlocks scalable deployments for real-time multimodal applications.

## 1 Introduction

Vision-Language Large Models (VLLMs) (Liu et al., 2023; 2024a; Li et al., 2024a) have emerged as a cornerstone in multimodal artificial intelligence, enabling sophisticated understanding and reasoning over both visual and textual modalities. These models have demonstrated remarkable performance across a wide range of applications, including video understanding, visual question answering, and multimodal content generation. However, the increasing demand for high-resolution visual inputs has led to a dramatic surge in the number of visual tokens processed by VLLMs, posing severe challenges in terms of memory footprint and computational efficiency during inference.

The primary bottleneck for efficient execution of VLLMs stems from the autoregressive nature of large language models (LLMs) (Achiam et al., 2023; Alexandre et al., 2023; Meta, 2024), which leads to a continuous increase in KV cache size and a quadratic growth in computational complexity during the prefilling and decoding stages. The KV cache, essential for maintaining past key-value pairs to accelerate autoregressive generation (Li et al., 2024b), becomes a critical resource bottleneck, especially when handling long visual sequences.

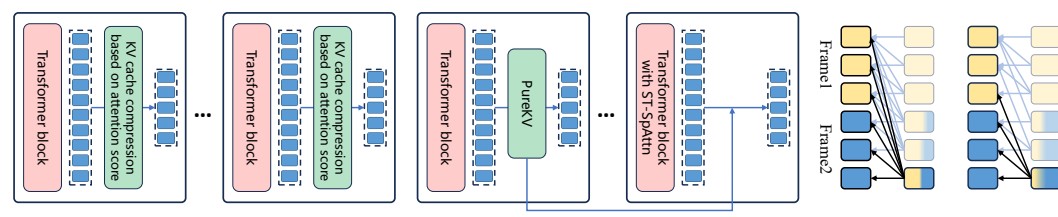

(a) KV cache compression based on attention score     (b) PureKV: Cross-Layer Importance Estimation     (c) Dense Attention    (d) ST-SpAttn

Figure 1: (a) The traditional KV cache compression method based on attention score calculates attention weights at each layer to evaluate tokens importance, which is not compatible with efficient attention mechanisms such as FlashAttention and Sparse Attention. (b) PureKV utilizes lower layer attention scores to identify critical KV cache in high layers, and is compatible with efficient attention mechanisms in the high layers, accelerating the prefilling stage. (c) Dense Attention leads to the gradual confusion of important and unimportant information at high layer. (d) ST-SpAttn generates cleaner and more structured KV, reducing noise while preserving key spatiotemporal dependencies.

To alleviate this issue, recent efforts (Tang et al.; Ge et al., 2023) have focused on compressing the KV cache by identifying and pruning less important tokens. While promising, these approaches (Zhang et al., 2023; Wan et al., 2024; Li et al., 2024e) typically rely on attention scores to estimate token importance — a strategy that inherently conflicts with modern efficient attention mechanisms such as FlashAttention (Dao et al., 2022; Dao, 2023; Shah et al., 2024) and Sparse Attention (Xu et al., 2025). These mechanisms, designed to accelerate attention computation, do not explicitly generate attention matrices, thereby rendering traditional attention-score-based pruning incompatible. Moreover, these methods fail to account for how sparse attention alters the information structure of the KV cache, which can affect the effectiveness of KV cache compression.

Recent works, such as StreamingLLM (Xiao et al., 2023) and window-based KV cache management (Beltagy et al., 2020; Han et al., 2023; Zuo et al., 2025), propose to compress the KV cache by retaining only the latest fixed length window or initial token, without this limitation. While effective in reducing memory consumption and enabling compatibility with efficient attention implementations, these methods lack the ability to dynamically preserve semantically critical tokens, often leading to the premature eviction of important information and consequent degradation in model performance.

In this paper, we tackle the fundamental challenge of **how to effectively identify and retain important KV cache entries while remaining fully compatible with efficient attention mechanisms**. In addition, we recognize that sparse attention can alter the information structure of KV cache. Therefore, we design a structured sparse pattern for video KV cache, which improves the effectiveness of KV cache compression strategy. The ultimate goal is to reduce memory usage in VLLMs and speed up the decoding stage while reducing the Time To First Token (TTFT) (Horton et al., 2024) in the prefilling stage.

To this end, we propose **PureKV**, a plug-and-play, efficient KV cache compression strategy that seamlessly integrates with modern attention accelerations. At the core of PureKV lies a lightweight KV cache importance estimator that leverages lower layer attention scores to approximate the importance of tokens in high layers. **Through statistical analysis, we show that lower layer attention scores serve as a sufficient statistic for estimating high-layer token importance.** This insight allows us to decouple the KV cache compression strategy from the computation of attention scores in high layers, thereby enabling full compatibility with efficient attention mechanisms.

In addition, as shown in Figure 1 (c), we find that dense attention (Vaswani et al., 2017) leads to the gradual confusion of important and unimportant information at high layers. This entanglement has a negative impact on the accuracy of high layer KV cache importance estimation. Therefore, we introduce a novel **Spatial-Temporal Sparse Attention (ST-SpAttn) mechanism** tailored to the inherent spatiotemporal redundancy in VLLMs. ST-SpAttn not only accelerates the prefill phase by exploiting spatial and temporal attention sparsity patterns, but also performs KV cache purification by suppressing background noise and redundant information across both spatial and temporal dimensions. Specifically, for spatial sparsity, we retain attention links to the first frame and within the current frame; for temporal sparsity, we preserve attention to the first frame and to the corre-

sponding tokens in the previous frame. This dual-path design ensures that only the most salient and temporally coherent tokens are preserved in the KV cache.

Our contributions are summarized as follows:

- We propose a lightweight, attention-score-free token importance estimator compatible with efficient attention mechanisms, enabling effective KV cache pruning without sacrificing generation quality.

- We design a Spatial-Temporal Sparse Attention mechanism that not only accelerates prefill computation but also purifies the KV cache, enhancing the effectiveness of cache eviction policies.

- We conduct extensive experiments on large-scale VLLMs, including VideoL-LaMA2 (Cheng et al., 2024) and Qwen2.5-VL (Bai et al., 2025), demonstrating that PureKV significantly reduces memory consumption and first-token latency while maintaining competitive performance across various video understanding tasks.

Our work bridges the gap between KV cache compression and efficient attention computation in VLLMs, offering a practical and scalable solution for real-world deployment of VLLMs under resource-constrained environments.

## 2 RELATED WORK

VLLMs (Li et al., 2023; 2024d; Xu et al., 2024) have emerged as a cornerstone of multimodal AI, unifying visual and linguistic modalities within a shared semantic space to enable cross-modal understanding and reasoning. Recent advanced models include LLaVA (Li et al., 2024a), which pioneers a lightweight "vision-as-language" interface via visual token projection; VideoL-LaMA2 (Cheng et al., 2024), which extends temporal modeling with spatiotemporal attention for long-form video understanding; and Qwen2.5-VL (Bai et al., 2025), a scalable framework featuring dynamic resolution adaptation, fine-grained spatial perception, and built-in visual agent capabilities. These models exemplify the trend toward greater generality, higher input fidelity, and broader functional integration. To address the growing computational demands of such models, efficient inference techniques have become critical. Approaches include quantization (Abreu et al., 2025; Tan et al., 2024), which reduces parameter precision with minimal accuracy loss; KV cache optimization (Kwon et al., 2023; Feng et al., 2024; Liu et al., 2024b; Ge et al., 2023), which improves memory efficiency during autoregressive generation; and system-level innovations such as FlashAttention-2 (Dao, 2023) and asynchronous scheduling in frameworks like vLLM (Kwon et al., 2023), which significantly accelerate end-to-end throughput.

Autoregressive decoding in LLMs (Touvron et al., 2023) faces severe memory bottlenecks due to the linear growth of the KV cache with sequence length (Shi et al., 2024; Feng et al., 2025), which requires efficient KV cache management for long-context inference. Existing methods (Tu et al., 2024; Wan et al., 2024) primarily fall into attention-score-based (He et al., 2024; Liu et al., 2024c) pruning and window-based (Xiao et al., 2023) retention, each addressing memory overhead but with different trade-offs. Traditional pruning techniques like $H_2O$ (Zhang et al., 2023) and SnapKV (Li et al., 2024e) dynamically evict KV cache by computing per-layer attention scores, yet their dependency on full attention matrices renders them incompatible with hardware-optimized kernels such as FlashAttention and Sparse Attention (Roy et al., 2021; Lou et al., 2024), diminishing their practical utility. In contrast, StreamingLLM (Xiao et al., 2023) uses a fixed-length sliding window strategy, retaining only initial "attention sink" tokens and recent tokens, thereby avoiding score-calculation and ensuring compatibility with modern accelerators. Despite reducing memory consumption and enhancing system compatibility, these windowed approaches suffer from static retention policies that indiscriminately evict tokens beyond the window, often discarding semantically critical information. Hybrid solutions like adaptive budget allocation (Feng et al., 2024) attempt to dynamically adjust compression per attention head, yet introduce latency overheads that negate efficiency gains. Consequently, the compatibility of dynamic KV cache importance recognition with efficient attention mechanisms has not been resolved, which has prompted the proposal of PureKV.

Self-Attention (Vaswani et al., 2017), while foundational to Transformer success, suffer from quadratic computational and memory complexity with respect to sequence length. Sparse Atten-

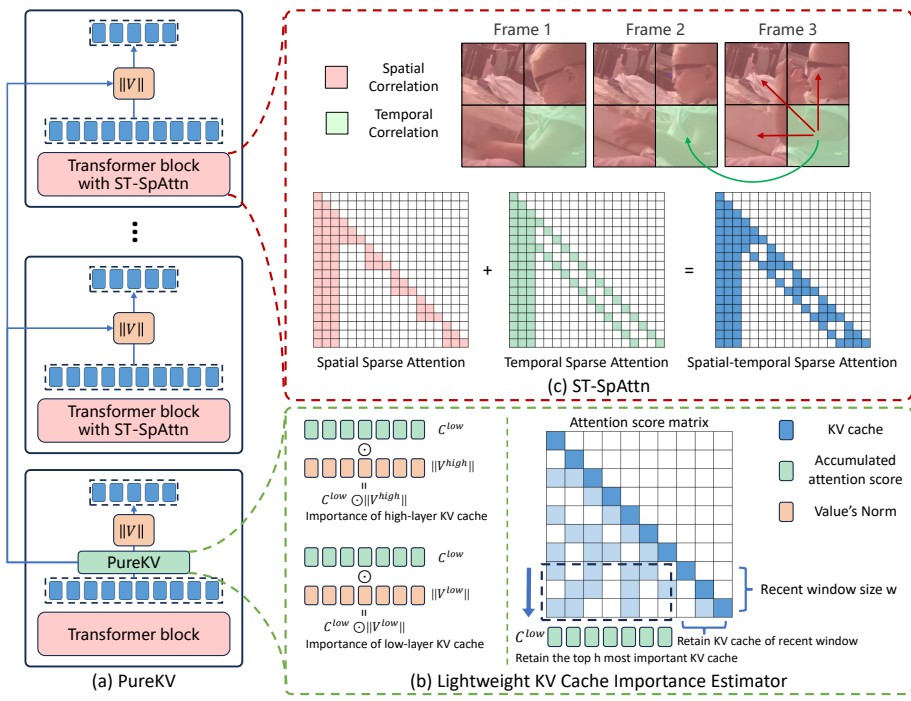

Figure 2: Overview of our PureKV method. PureKV is a plug-and-play framework for KV cache optimization, compatible with efficient attention mechanisms. PureKV introduces a lightweight importance estimator that utilizes layer attention scores and the L2 norm of high V vectors to estimate KV cache importance, avoiding explicit computation of high attention. By combining Spatial-Temporal Sparse Attention, PureKV suppresses background noise and irrelevant visual interference, eliminates redundancy in consecutive frames, and the resulting purified KV cache significantly improves the accuracy and robustness of subsequent KV cache compression strategies.

tion (Tay et al., 2020; Yuan et al., 2025; Xi et al., 2025) addresses this bottleneck by selectively computing attention scores over a subset of token pairs, thereby reducing complexity to sub-quadratic or even linear scales. Recent innovations include fixed-pattern sparsity (e.g., sliding window in Longformer (Beltagy et al., 2020), block-sparsity in Sparse Transformer (Child et al., 2019)), content-aware approximations (e.g., LSH-based Reformer (Chen et al., 2020), clustering in Routing Transformer (Roy et al., 2021)), and kernelized methods (e.g., Linear Transformer (Wang et al., 2020)). Hybrid approaches like BigBird (Zaheer et al., 2020) further combine local, random, and global attention to theoretically preserve expressiveness. However, these methods predominantly focus on optimizing training-time efficiency and long-context modeling capabilities. While they effectively reduce FLOPs and memory footprints during forward passes, their implications for inference-time optimizations—particularly the interaction with KV cache compression algorithms—remain largely unexplored. No existing sparse attention scheme explicitly designs sparsity patterns to enhance or synergize with dynamic KV cache pruning, quantization, or eviction strategies.

## 3 METHOD

### 3.1 LIGHTWEIGHT KV CACHE IMPORTANCE ESTIMATOR

Traditional attention-score-based KV cache compression methods rely on computing attention weights at every layer to assess token importance, which inherently prevents compatibility with highly optimized attention implementations such as FlashAttention and Sparse Attention——these efficient kernels do not expose or compute explicit attention matrices during inference. To overcome this limitation, we propose a lightweight KV cache importance estimator that can accurately estimate KV cache importance while maintaining compatibility with state-of-the-art attention accelerators.

We have demonstrated through experiments and statistical analysis that the lower layer attention scores of VLLMs can serve as effective proxies for estimating the cache importance of high layer KV cache. Inspired by this observation, our method, PureKV, computes attention scores only in the lower layer and leverages them to estimate the importance of KV cache in subsequent layers, thereby enabling the integration of efficient attention mechanisms in high layers.

Given an input sequence of length $l$, for lower layers, we retain a recent window of size $w$ and additionally retain the top h most important KV cache in non-recent segment. To estimate the importance of KV cache, we calculate the attention score matrix:

$$A^{low} = softmax(\frac{QK^T}{\sqrt{d_k}}),\tag{1}$$

where $Q$ denotes the query matrix of the input tokens, $d_k$ is the dimension of K. Most existing methods (Zhang et al., 2023; Li et al., 2024e) rely on accumulated attention scores to identify important tokens. However, due to the lower-triangular structure of the attention matrix, such approaches inherently bias toward earlier tokens (He et al., 2024). As illustrated in Figure 2 (b), to mitigate this bias, PureKV computes the cumulative attention score of tokens in the recent window with respect to those in the non-recent segments:

$$C^{low} = \sum_{i=l-w}^{i<l} A_{i,j}, 0 \le j < l-w.\tag{2}$$

Previous work mainly rely on attention based metrics to evaluate the importance of KV cache, ignoring the impact of V vectors on output. As shown in Figure 3, the size of the V significantly affects the output of the attention mechanism. To consider the influence of V, we weight the accumulated attention scores based on the L2 norm of the corresponding V vector:

$$S^{low} = C^{low} \odot \left\| V_{0:l-w}^{low} \right\|.\tag{3}$$

where $S^{low}$ denotes the final importance score for tokens in the non-recent segment. We retain the recent $w$ tokens (recent window) and select the top h most important tokens from the non-recent segment based on $S^{low}$.

In high layers, we similarly retain a recent window of size w, as well as the top h most important KV cache from non-recent segment. For high layers that adopt efficient attention mechanisms, PureKV reuses the lower layer accumulated attention score $C^{low}$ and calculates the L2 norm of V vector to estimate KV cache importance:

$$\hat{S}^{high} = C^{low} \odot \left\| V_{0:l-w}^{high} \right\|.\tag{4}$$

This cross-layer importance estimation enables accurate KV cache selection without computing attention scores in high layers, thus preserving full compatibility with FlashAttention and other high-performance attention backends.

## 3.2 SPATIAL-TEMPORAL SPARSE ATTENTION

Although previous sparse attention mechanisms accelerated the prefilling stage by only focusing on a portion of the input elements during computation, they often alter the information structure of the generated KV cache, potentially degrading the effectiveness of downstream KV cache selection and compression. Existing KV cache pruning strategies fail to account for such structural modifications, leading to suboptimal performance. Due to causal attention, the KV cache at position $j$ in layer $i$ aggregates information from the first $j + 1$ tokens in layer $i - 1$, resulting in progressive mixing of important and unimportant information. This entanglement adversely impacts the accuracy of importance estimation in high layers.

To mitigate this issue, we propose Spatial-Temporal Sparse Attention (ST-SpAttn), designed to purify the KV cache by disentangling informative signals from spatial and temporal noise. As illustrated in Figure 2 (c), ST-SpAttn consists of two components—spatial and temporal sparsity—specifically tailored for video understanding tasks.

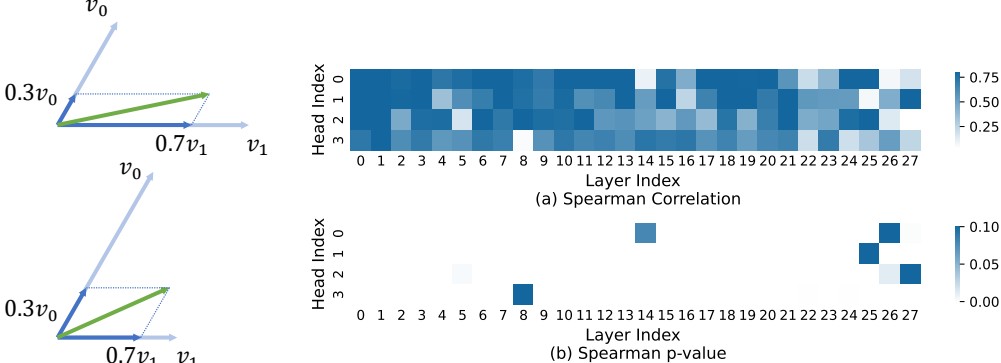

Figure 3: Under fixed attention weight conditions, the size of the V vector also significantly affects the output results of the attention mechanism.

Figure 4: Cross-Layer importance Estimation correlation analysis. The experiment shows that the high layer KV cache importance estimation based on lower layer attention scores is significantly positively correlated with the true high layer KV cache importance. (VideoLLaMA2 uses Group query attention, divides the heads into 4 groups, with each group sharing KV cache.)

For spatial purification, we employ spatial sparse attention by preserving only two types of interactions: (1) attention between each token and the first frame (capturing long-range visual consistency), and (2) intra-frame attention within the current frame (retaining local spatial context). This effectively suppresses background noise and irrelevant visual distractions.

For temporal purification, we introduce temporal sparsity by retaining each token's attention only to its corresponding token in the previous frame, enhancing temporal coherence, while maintaining connections to the first frame for global context anchoring. This reduces redundancy from highly similar consecutive frames.

Overall, this dual path sparsity generates cleaner and more structured KV cache, reduces noise, accelerates the prefilling stage of VLLMs, while preserving key spatiotemporal dependencies. The resulting purified KV cache significantly improves the accuracy and robustness of subsequent compression strategies.

### 3.3 STATISTICAL VALIDATION OF CROSS-LAYER IMPORTANCE ESTIMATION

To verify PureKV's core hypothesis that KV cache importance in high layers can be effectively estimated using lower layer attention score, we formalized the following hypothesis: the KV cache importance ranking obtained by weighting lower layer cumulative attention scores with L2 norm of high layer V is similar to the ranking obtained by weighting high layer cumulative attention scores with L2 norm of high layer V. Specifically, our goal is to prove:

$$rank(\hat{S}^{high}) \approx rank(S^{high}), \tag{5}$$

$$S^{high} = C^{high} \odot \left\| V_{0:l-w}^{high} \right\|, \tag{6}$$

where $rank(S) = \{r_{s_0}, r_{s_1}, ... r_{s_{l-w-1}}\}$, $r_{s_j}$ denotes the rank of $s_j$ in sequence $S$.

To quantify the agreement between these two rankings, we employ Spearman's rank correlation coefficient (Sedgwick, 2014), which measures the monotonic relationship between two ranked variables:

$$\rho(\hat{S}^{high}, S^{high}) = 1 - \frac{6\sum_{j=0}^{n}(r_{\hat{s}_j^{high}} - r_{s_j^{high}})^2}{n(n^2 - 1)}, \tag{7}$$

where $n = l - w$. The coefficient $\rho \in [-1, 1]$, with values closer to 1 indicating strong rank consistency.

We use the cumulative attention scores from the layer 1 to compute the estimated importance scores $\hat{S}^{high}$ for high layers, and compute the Spearman rank correlation between $\hat{S}^{high}$ and the ground-truth scores $S^{high}$ (computed using the respective layer's own attention). As shown in Figure 4,

Table 1: Performance of KV cache compression strategy based on Qwen2.5-VL-7B on MVBench. **The best results** are highlighted in bold. The second result is highlighted with an underline.

| | AS | AP | UA | OI | OS | AL | ST | SC | CO | CI | Avg. |
|---|---|---|---|---|---|---|---|---|---|---|---|
| Full Cache | 0.7215 | 0.6995 | 0.5559 | 0.7014 | 0.1674 | 0.5052 | 0.7249 | 0.2859 | 0.3145 | 0.4595 | 0.5136 |
| *20% Cache Budget* | | | | | | | | | | | |
| H$_2$O | 0.6919 | 0.6812 | 0.4202 | 0.7025 | 0.1851 | 0.4587 | 0.6873 | 0.2672 | 0.3151 | 0.2852 | 0.4695 |
| SnapKV | 0.6838 | 0.6933 | 0.4781 | 0.7153 | 0.1799 | 0.4595 | 0.7000 | 0.2924 | 0.2934 | 0.3124 | 0.4808 |
| StreamingLLm | 0.6665 | 0.7005 | 0.5056 | 0.7563 | 0.1867 | 0.4675 | **0.7519** | 0.2046 | 0.3216 | 0.3802 | 0.4941 |
| FastV | 0.7273 | 0.6973 | 0.5581 | 0.7127 | 0.1742 | 0.5111 | 0.7002 | 0.3095 | **0.3274** | 0.4366 | 0.5154 |
| LOOK-M | 0.0074 | 0.0126 | 0.1268 | 0.0187 | 0.0014 | 0.0948 | 0.0292 | 0.0267 | 0.0000 | 0.2929 | 0.0610 |
| PureKV | **0.7490** | **0.7142** | **0.5624** | **0.7956** | **0.1957** | **0.5562** | 0.7242 | **0.3488** | 0.3086 | **0.4749** | **0.5429** |
| *10% Cache Budget* | | | | | | | | | | | |
| H$_2$O | 0.6794 | 0.6715 | 0.3488 | 0.7145 | 0.1594 | 0.4751 | 0.5982 | 0.2005 | 0.2278 | 0.1586 | 0.4234 |
| SnapKV | 0.7041 | 0.6726 | 0.4180 | 0.7253 | 0.1869 | 0.4570 | 0.6854 | 0.2554 | 0.2421 | 0.1579 | 0.4505 |
| StreamingLLm | 0.6790 | 0.7001 | 0.4835 | 0.7645 | 0.2263 | 0.4689 | 0.7637 | 0.1915 | 0.2238 | 0.2996 | 0.4801 |
| FastV | 0.7210 | 0.6896 | **0.5826** | 0.7266 | 0.1979 | 0.5086 | 0.7093 | 0.2588 | 0.2558 | **0.4219** | 0.5072 |
| LOOK-M | 0.0097 | 0.0119 | 0.0722 | 0.0167 | 0.0014 | 0.0913 | 0.0268 | 0.0283 | 0.0000 | 0.1309 | 0.0389 |
| PureKV | **0.7399** | **0.7167** | 0.4999 | **0.7865** | **0.2281** | **0.6091** | **0.7902** | **0.3760** | **0.2847** | 0.3289 | **0.5360** |

across most layers, the Spearman correlation exceeds 0.4, and even in the highest layers, the majority of correlations remain above 0.2. Furthermore, the correlations are statistically significant ($p < 0.05$) in most cases, indicating a significant positive rank agreement between $\hat{S}^{high}$ and $S^{high}$.

These results provide strong empirical support for the validity of our cross-layer importance estimation strategy: **despite not computing attention in high layers, PureKV can reliably identify important KV cache based on lower layer attention score and L2 norm of high layer V.**

## 4 EXPERIMENTS

### 4.1 SETTING

MVBench (Li et al., 2024c) is a comprehensive multimodal video understanding benchmark that covers 20 challenging video understanding tasks. We extracted 14 tasks from them to validate our algorithm, namely: Action Sequence (AS), Action Prediction (AP), Unexpected Action (UA), Object Interaction (OI), Object Shuffle (OS), Action Localization (AL), Scene Transition (ST), Action Count (AC), State Change (SC), Object Existenc (OE), Moving Count (MC), Moving Attribute (MA), Egocentric Navigation (EN), Counterfactual Inference (CI). We use ROUGE as the experimental evaluation metric.

To evaluate PureKV, we conduct extensive experiments on two advanced VLLMs: VideoL-LaMA2 (Cheng et al., 2024) and Qwen2.5-VL-7B (Bai et al., 2025). We compare against five representative KV cache compression strategies: H$_2$O (Zhang et al., 2023), SnapKV (Li et al., 2024e), and StreamingLLM (Xiao et al., 2023), which are originally designed for text-based scenarios, as well as FastV (Chen et al., 2024) and LOOK-M (Wan et al., 2024), tailored for visual tasks. We conduct experiments on a NVIDIA A100 with 40GB memory.

### 4.2 MAIN EXPERIMENT RESULTS

We conducted a comprehensive evaluation of PureKV in video understanding scenarios. As shown in Table 1 and 2, PureKV achieved efficient memory reduction while maintaining strong task performance within budget constraints. Specifically, compared to full cache, PureKV reduces KV cache memory usage by 80% with only a slight decrease in performance, demonstrating PureKV's ability to significantly reduce memory usage with minimal performance cost.

Moreover, PureKV outperforms other baseline methods in most video understanding tasks. Previous methods typically relied solely on attention scores to evaluate KV cache importance, while ignoring the impact of the V vector on the final output of the attention mechanism. This has led to biased estimates of importance. In contrast, PureKV obtains a more accurate and robust importance score

Table 2: Performance of KV cache compression strategy based on VideoLLaMA2 on MVBench. **The best results** are highlighted in bold. The second result is highlighted with an underline.

|  | AS | AP | UA | AC | SC | OE | MC | MA | EN | CI | Avg. |
|---|---|---|---|---|---|---|---|---|---|---|---|
| Full Cache | 0.7676 | 0.6883 | 0.7851 | 0.5000 | 0.6869 | 0.6600 | 0.4750 | 0.6800 | 0.6117 | 0.7805 | 0.6635 |
| *20% Cache Budget* | | | | | | | | | | | |
| H$_2$O | 0.5947 | 0.5388 | 0.3204 | 0.3500 | 0.1752 | 0.1350 | 0.2426 | 0.0552 | 0.4381 | 0.2342 | 0.3084 |
| SnapKV | 0.7288 | 0.6375 | 0.5844 | 0.4846 | 0.4102 | 0.2919 | 0.2933 | 0.2673 | 0.4470 | 0.6660 | 0.4811 |
| StreamingLLm | **0.7588** | 0.6622 | 0.6916 | 0.4764 | 0.4409 | 0.3400 | 0.3104 | 0.3603 | 0.5053 | **0.7276** | 0.5274 |
| FastV | 0.7357 | 0.6667 | 0.617 | 0.4415 | 0.4488 | 0.3779 | 0.3322 | 0.3147 | 0.4201 | 0.7135 | 0.5068 |
| PureKV | **0.7588** | **0.6930** | **0.6975** | **0.4850** | **0.4751** | **0.7206** | **0.4650** | **0.5814** | **0.5875** | 0.7248 | **0.6189** |
| *10% Cache Budget* | | | | | | | | | | | |
| H$_2$O | 0.4015 | 0.4697 | 0.1406 | 0.3900 | 0.1224 | 0.1500 | 0.2852 | 0.0991 | 0.1531 | 0.2370 | 0.2449 |
| SnapKV | 0.5835 | 0.6055 | 0.3292 | 0.4000 | 0.2323 | 0.2332 | 0.3376 | 0.2558 | 0.2263 | 0.2970 | 0.3500 |
| StreamingLLm | 0.7133 | **0.6660** | 0.5033 | 0.3961 | 0.2994 | 0.2768 | 0.3391 | 0.3431 | 0.3810 | 0.5330 | 0.4451 |
| FastV | 0.3633 | 0.4814 | 0.2809 | 0.4495 | 0.2064 | 0.3469 | 0.2673 | 0.2369 | 0.2998 | 0.2603 | 0.3193 |
| PureKV | **0.7195** | 0.6643 | **0.5190** | **0.4800** | **0.3586** | **0.3792** | **0.4563** | **0.3897** | **0.4192** | **0.6610** | **0.5047** |

Table 3: Inference speed based on VideoL-LaMA2.

| Method | Budget | Prefilling Latency | Decoding Latency |
|---|---|---|---|
| Full Cache | 100% | 0.1190 ms/token | 36.73 ms/token |
| PureKV | 50% | 0.0366 ms/token | 31.87 ms/token |
|  | 35% | 0.0370 ms/token | 28.50 ms/token |
|  | 20% | 0.0376 ms/token | 28.32 ms/token |
|  | 5% | **0.0355 ms/token** | **27.92 ms/token** |

Table 4: Ablation study. CLIE: Cross-Layer Importance Estimation, ST-SpAttn: Spatial-Temporal Sparse Attention, V: Weighted with L2 norm of V.

| CLIE | ST-SpAttn | V | Qwen2.5-VL | VideoLLaMA2 |
|---|---|---|---|---|
| ✗ | ✗ | ✔ | 0.7307 | 0.6985 |
| ✔ | ✗ | ✔ | 0.7311 | 0.7020 |
| ✔ | ✔ | ✗ | 0.7212 | 0.6936 |
| ✔ | ✔ | ✔ | **0.7490** | **0.7588** |

by weighting the accumulated attention score with the L2 norm of the corresponding V vector, taking into account the influence of V. This design reduces the estimation bias of KV cache importance and improves the compression efficiency of KV cache.

Furthermore, PureKV combines Spatial-Temporal Sparse Attention in the prefilling stage to purify KV cache. By suppressing spatial background noise and temporal redundancy, this purification produces KV states with clearer information structures. The resulting structured KV cache improves the fidelity, accuracy, and robustness of subsequent KV compression algorithms. PureKV performs well in multiple complex video understanding tasks, verifying its effectiveness and universality.

As shown in Table 3, while leading in accuracy, PureKV utilizes layer attention scores to estimate the importance of high layers KV cache, reducing the computational cost of importance estimation and achieving faster decoding speed than the comparison method. In addition, by seamlessly integrating with efficient attention mechanisms, PureKV accelerates the prefilling phase and reduces the TTFT.

### 4.3 INFLUENCE OF VARIOUS CACHE BUDGETS

To evaluate the effectiveness of PureKV under different cache budgets, we conducted experiments based on VideoLLaMA2 and Qwen2.5-VL. The results are presented in Table 1 and 2, respectively. As the cache budget decreases, the performance of the other KV cache compression strategies has significantly declined. In contrast, PureKV demonstrates excellent robustness and efficiency. It's noted that under strict memory limitations — only 10% of KV cache is retained - PureKV maintains stable performance on both VLLMs. This highlights PureKV's ability to accurately identify and retain critical information, minimizing context loss while significantly reducing memory usage.

### 4.4 ABLATION STUDY

To analyze the contribution of each component in PureKV, we conducted ablation studies on two representative VLLMs: Qwen2.5-VL and VideoLLaMA2. Table 4 presents the results of two VLLMs on the Action Sequence (AC) task, where we evaluated the impact of three key design choices:

Cross-Layer Importance Estimation (CLIE), Spatial-Temporal Sparse Attention (ST-SpAttn), and Weighting with L2 norm of V.

**Impact of Cross-Layer Importance Estimation.** Table 4 shows that CLIE does not cause performance degradation in Qwen2.5-VL and VideoLLaMA2. This indicates that we can use lower layer attention scores to estimate the importance of high layer KV cache.

**Impact of Spatial-Temporal Sparse Attention.** When ST-SpAttn is disabled, the performance of Qwen2.5-VL and VideoLLaMA2 significantly decreases. This highlights the importance of Spatial-Temporal purification in suppressing irrelevant visual interference and temporal redundancy. ST-SpAttn ensures that the information structure of KV cache is clearer, thereby improving the quality of compressed representation.

**Effect of V Vector Weighting.** Disabling the weighting with the L2 norm of V results in performance drops. This underscores the significance of incorporating V vector into KV cache importance scoring. By accounting for the contribution of V to the final attention output, PureKV achieves more accurate and robust KV cache prioritization, enhancing KV cache compression efficiency.

### 4.5 HYPERPARAMETER ANALYSIS

We conducted a hyperparameter analysis aimed at exploring the effects of CLIE and ST-SpAttn activation at different initiation layers on PureKV performance. As shown in Figure 5, when the CLIE Layer Index is set to 0, the performance of PureKV significantly decreases, indicating that the initial layer may lack sufficient contextual information for effective cross-layer importance estimation. When the CLIE layer index is 2, PureKV performs the best, and VLLMs can use efficient attention mechanisms in high layers to accelerate prefilling inference.

In addition, Figure 5 also reveals that ST-SpAttn typically leads to better performance when activated in high layers, but this advantage does not increase indefinitely with depth. The high layer KV cache mixes important and unimportant information, which is more suitable for spatiotemporal filtering. By applying ST-SpAttn in high layers, irrelevant visual interference and temporal redundancy can be effectively suppressed, ensuring that KV cache only retains more refined and structured information, ultimately improving the quality of compressed representations.

Figure 5: CLIE Layer Index: the lower layer index used to estimate importance of high layer KV cache. ST-SpAttn Layer Index: the layer index at which SpAttn is activated. Since ST-SpAttn does not explicitly calculate attention score, the ST-SpAttn Layer Index is greater than the CLIE Layer Index.

## 5 CONCLUSION

PureKV is a plug-and-play KV cache compression framework compatible with efficient attention mechanisms. It introduces a cross-layer importance estimation (CLIE) strategy that leverages attention scores from lower layer and V vectors from high layers to estimate the importance of KV caches in high layers, with its validity supported by experiments and statistical analysis. In addition, we found that dense attention leads to the gradual confusion of important and unimportant information at high levels, thereby reducing the accuracy of importance estimation. To address this issue, we propose a novel Spatial-Temporal Sparse Attention (ST-SpAttn) mechanism that purifies KV cache by suppressing spatial background noise and temporal redundancy. This significantly improves the accuracy and robustness of KV cache compression. Extensive experiments on multiple VLLMs and video understanding tasks have demonstrated the effectiveness of PureKV.

## 6 ETHICS STATEMENT

This work follows the ethical principles outlined in the ICLR Code of Ethics, emphasizing responsible management, scientific excellence, and social well-being. We acknowledge global stakeholders in machine learning research and strive to ensure that our contributions benefit society while minimizing potential harm. Our research adheres to high standards of integrity, transparency, and reproducibility, and reports methods and results accurately and honestly. We carefully considered the broader impact of our work, including potential risks to privacy, security, and fairness, and collaborated with experts in relevant fields to mitigate unintended consequences. Any data used in this study has been processed in accordance with ethical approval, respecting privacy and confidentiality. We are committed to promoting inclusivity, avoiding discrimination, and ensuring that our research results are easily accessible and socially responsible.

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

# 7 APPENDIX

## 7.1 THE USE OF LARGE LANGUAGE MODELS

In the preparation of this manuscript, LLMs is utilized as a general-purpose assist tool for specific tasks. The LLMs is employed solely for the following purposes:

- Spelling and Grammar Checking: The LLMs is used to identify and correct spelling errors and grammatical inconsistencies, such as verb tense agreement, across the manuscript.
- Sentence Polishing: The LLMs provides suggestions for rephrasing sentences to enhance clarity and readability, without altering the original meaning or technical content of the text. All suggestions are reviewed and approved by the authors to ensure alignment with the intended scientific contributions.

The use of the LLMs is limited to these auxiliary tasks and does not contribute to the research ideation, methodology, analysis, or core writing of the paper. All scientific content, including ideas, arguments, and conclusions, is developed and written by the authors.

## 7.2 INTEGRATION WITH EXISTING KV CACHE COMPRESSION ALGORITHMS

To further validate the effectiveness of our proposed Cross-Layer Importance Estimation (CLIE) and Spatial-Temporal Sparse Attention (ST-SpAttn) mechanisms, we integrate them into several state-of-the-art KV cache compression algorithms. The results are summarized in Table 6, which compares the performance of these methods both independently and when augmented with PureKV.The table 6 demonstrates that integrating PureKV significantly boosts the performance of existing KV cache compression algorithms across various metrics.

The consistent performance gains across different algorithms and metrics underscore the generalizability and robustness of our proposed mechanisms. CLIE and ST-SpAttn can be seamlessly integrated into various KV cache compression frameworks, enhancing their efficiency and effectiveness without requiring extensive modifications.

Table 5: Applying PureKV in Audio-Video LLMs.

| | AVSD |
|---|---|
| Full Cache | 0.4795 |
| $H_2O$ | 0.3527 |
| SnapKV | 0.4249 |
| StreamingLLM | 0.4249 |
| FastV | 0.4224 |
| PureKV | **0.4265** |

## 7.3 APPLYING PUREKV IN AUDIO-VIDEO LLMS.

To further validate the effectiveness and versatility of our proposed PureKV algorithm, we conducted experiments using the Audio Visual Scene Aware Dialogue (AVSD) dataset. The AVSD dataset focuses on dialogue understanding tasks and provides rich audiovisual information, making it an ideal benchmark to evaluate the performance of Audio-Video large language models (AV-LLMs). Table 5 presents the performance of various KV cache compression algorithms on the AVSD dataset. Our proposed PureKV method significantly outperforms all other baselines.

Our experiments on the AVSD dataset validate the effectiveness and versatility of PureKV. By integrating CLIE and ST-SpAttn mechanisms, PureKV achieves superior performance compared to existing methods, demonstrating its potential to enhance the efficiency and accuracy of audio-video LLMs in dialogue understanding tasks.

## 7.4 EXPERIMENTAL ANALYSIS OF DIFFERENT SPARSE ATTENTION

To evaluate the effectiveness of different sparse attention mechanisms within the PureKV framework, we conduct a comprehensive ablation study on VideoLLaMA2. As shown in Figure 6, we compare five sparse attention: Atrous Attention, Local Attention, Spatial Sparse Attention, Temporal Sparse Attention, and our proposed Spatial-Temporal Sparse Attention (ST-SpAttn).

As shown in Table 7, our proposed Spatial-Temporal Sparse Attention (ST-SpAttn) strikes an optimal balance between spatial and temporal modeling. It achieves the best overall performance with an average accuracy of 0.6189, outperforming all other sparse variants. These results confirm that ST-SpAttn, as integrated into PureKV, effectively enhances both efficiency and accuracy by leveraging structured sparsity that aligns with the intrinsic structure of video data.

Table 6: VideoLLaMA2: Combining other KV cache compression algorithms with PureKV. **The best results** are highlighted in bold.

| | AC | AP | UA | AC | SC | OE | MC | MA | EN | CI | Avg. |
|---|---|---|---|---|---|---|---|---|---|---|---|
| Full Cache | 0.7676 | 0.6883 | 0.7851 | 0.5000 | 0.6869 | 0.6600 | 0.4750 | 0.6800 | 0.6117 | 0.7805 | 0.6635 |
| $H_2O$ | 0.5947 | 0.5388 | 0.3204 | 0.3500 | 0.1752 | 0.1350 | 0.2426 | 0.0552 | 0.4381 | 0.2342 | 0.3084 |
| *+PureKV* | **0.6796** | **0.6818** | **0.5202** | **0.4800** | **0.3792** | **0.4221** | **0.4150** | **0.2665** | **0.4530** | **0.6529** | **0.4950** |
| SnapKV | 0.7288 | 0.6375 | 0.5844 | **0.4846** | 0.4102 | 0.2919 | 0.2933 | 0.2673 | 0.4470 | 0.6660 | 0.4811 |
| *+PureKV* | **0.7538** | **0.6758** | **0.6381** | 0.4817 | **0.4492** | **0.4774** | **0.4717** | **0.4686** | **0.5671** | **0.6967** | **0.5680** |
| StreamingLLM | **0.7588** | 0.6622 | **0.6916** | 0.4764 | 0.4409 | 0.3400 | 0.3104 | 0.3603 | 0.5053 | 0.7276 | 0.5274 |
| *+PureKV* | 0.7556 | **0.6771** | 0.6833 | **0.5050** | **0.4659** | **0.4804** | **0.4500** | **0.3750** | **0.5105** | **0.7313** | **0.5635** |
| FastV | 0.7357 | 0.6667 | 0.6170 | 0.4415 | 0.4488 | 0.3779 | 0.3322 | 0.3147 | 0.4201 | 0.7135 | 0.5068 |
| *+PureKV* | **0.7368** | **0.6846** | **0.6190** | **0.5000** | **0.4906** | **0.4541** | **0.4500** | **0.3345** | **0.5539** | **0.7177** | **0.5541** |

Table 7: VideoLLaMA2: Purekv applies different sparse attention. **The best results** are highlighted in bold. The second result is highlighted with an underline.

| | AC | AP | UA | AC | SC | OE | MC | MA | EN | CI | Avg. |
|---|---|---|---|---|---|---|---|---|---|---|---|
| Full Cache | 0.7676 | 0.6883 | 0.7851 | 0.5000 | 0.6869 | 0.6600 | 0.4750 | 0.6800 | 0.6117 | 0.7805 | 0.6635 |
| Atrous Attention | 0.5103 | 0.4939 | 0.3390 | 0.2900 | 0.4354 | 0.4700 | 0.3256 | 0.2054 | 0.2766 | 0.4479 | 0.3794 |
| Local Attention | 0.7310 | 0.6671 | 0.6607 | 0.5325 | 0.3861 | 0.3265 | 0.2498 | 0.2403 | 0.4900 | 0.7113 | 0.4995 |
| Spatial Sparse Attention | 0.7464 | 0.6617 | 0.6723 | **0.5329** | **0.6219** | 0.3420 | 0.3059 | 0.3727 | 0.5160 | **0.7327** | 0.5504 |
| Temporal Sparse Attention | 0.7002 | 0.6315 | 0.6253 | 0.2757 | 0.5540 | 0.5750 | 0.3800 | 0.3629 | 0.5002 | 0.6976 | 0.5302 |
| Spatial-Temporal Sparse Attention | **0.7588** | **0.6930** | **0.6975** | 0.4850 | 0.4751 | **0.7206** | **0.4650** | **0.5814** | **0.5875** | 0.7248 | **0.6189** |

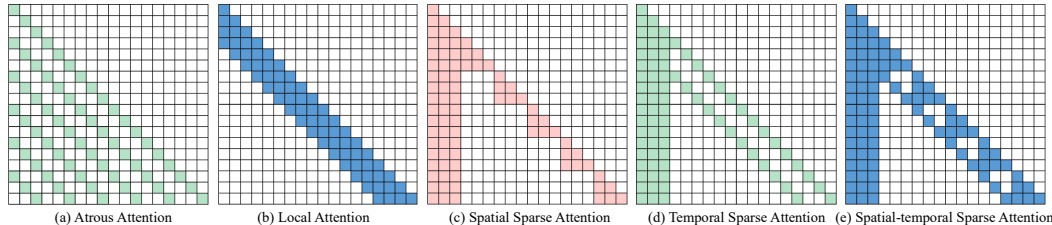

(a) Atrous Attention    (b) Local Attention    (c) Spatial Sparse Attention    (d) Temporal Sparse Attention    (e) Spatial-temporal Sparse Attention

Figure 6: Different Sparse Attention.

