# OpenReview forum: "PureKV: Plug-and-Play KV Cache Optimization with Spatial-Temporal Sparse Attention for Vision-Language Large Models"
_ICLR.cc/2026/Conference — ICLR 2026 Conference Withdrawn Submission_

### Official Review · Reviewer_xK7L · 2025-10-15

**Soundness:** 2
**Presentation:** 3
**Contribution:** 2
**Rating:** 2
**Confidence:** 4

**Summary:**

This paper argues that previous sparse attention and KV cache methods are incompatible with the FlashAtten kernel and largely static. To address this, the authors propose a dedicated sparse attention mechanism that purifies spatial noise and reduces temporal redundancy in the KV cache.

**Strengths:**

The decomposition of spatial and temporal redundancy is good.

**Weaknesses:**

I have several concerns regarding the claims and evaluations in this paper:

1. Adaptability of attention-based methods
The authors claim that existing attention-based methods cannot adapt to FlashAtten. However, this statement is not entirely accurate. As far as I know, several existing approaches employ block-wise probing strategies that efficiently obtain attention scores. Examples include NSA, MoBA, SeerAttention, and ZipVL.

[1] Hardware-Aligned and Natively Trainable Sparse Attention

[2] MoBA: Mixture of Block Attention for Long-Context LLMs

[3] SeerAttention: Learning Intrinsic Sparse Attention in Your LLMs

[4] ZipVL: Efficient Large Vision-Language Models with Dynamic Token Sparsification

2. Static vs. dynamic sparsity
The paper further claims that current methods are static. This characterization is misleading, as it mainly applies to Top-k based methods. In contrast, Top-p based dynamic KV methods have already been explored, such as Twilight and ZipVL, which demonstrate dynamic sparsity in practice.

[1] Twilight: Adaptive Attention Sparsity with Hierarchical Top-p Pruning

[2] ZipVL: Efficient Large Vision-Language Models with Dynamic Token Sparsification

3. Lack of comprehensive comparisons
The experimental evaluation is insufficient. The method is only compared against StreamingLLM and H2O, which does not establish a strong case for state-of-the-art performance. More comprehensive benchmarks should be included, such as VideoMME, LongVideoBench, PerceptionTest, ActNet-QA, and NextQA. In addition, evaluation on prolonged video cases exceeding 20k context length is necessary. More extensive experiments on LLaVA-Video are also expected.

4. Questionable necessity of cross-layer estimation
The proposed cross-layer estimation strategy seems unnecessary. Existing methods (e.g., SeerAttention) already adopt fine-grained token selection to skip computation without discarding tokens. This avoids excluding potentially important information, whereas cross-layer dropping may risk compounding information loss from earlier layers.

**Questions:**

Please refer to the weakness

---

> ### Author Response · Authors · 2025-12-04
> **[1/3] Official Comment by Authors**
>
> # Weaknesses 1
> Thank you for mentioning recent work such as NSA, MoBA, SeerAttention, and ZipVL. We attach great importance to this feedback and take this opportunity to more accurately define PureKV's technological positioning and contribution boundaries.
>
> Firstly, we would like to clarify that this paper focuses on train free, plug and play KV cache compression method.Its core goal is to achieve inference acceleration and memory optimization of VLLMs by optimizing KV cache without modifying model weights or introducing additional training.
>
> NSA, MoBA, SeerAttention are essentially learnable sparse attention mechanisms that introduce lightweight trainable modules to predict approximate importance scores on partitioned KV cache to guide sparse attention patterns. However, these methods:
> - Introduced additional training.
> - The output 'importance score' is not the original attention score, so it cannot be directly combined with KV compression strategies that rely on real attention weights such as H2O, SnapKV, LOOK-M, etc
> - The blocking strategy also limits the flexibility of KV cache selection, making it difficult to support dynamic and fine-grained token-level eviction.
>
> ZipVL is compatible with sparse attention through an additional attention score calculation, but still does not eliminate the computational overhead caused by calculating attention scores layer by layer.
>
> In contrast, the key breakthrough of PureKV is that calculating the true attention score only at the lower layers (such as layers 0-2) can provide reliable importance estimates for all higher layers. This makes:
> - High layers can be directly switched to FlashAttention/Sparse Attention.
> - Our framework can plug and play to enhance existing KV compression methods (we have shown in the appendix that integrating PureKV into H2O/SnapKV and other methods can also be compatible with efficient attention and improve performance);
> - No need for training, no need to modify the model.
>
> Therefore, the statement in our original text that "existing attention score based methods cannot adapt to FlashAttention" specifically refers to train free KV compression methods that rely on layer by layer explicit attention soccer (such as H2O, SnapKV, etc.), rather than referring to all attention optimization techniques. We have further clarified the terminology boundaries in the revised draft.
>
> Finally, we regret that we were unable to discuss and clarify technical details with the reviewers during this review process. Please kindly ask AC to consider this objective limitation in the comprehensive evaluation.
>
> Thanks again to the reviewer for the careful reading and constructive feedback!
>
> # Weaknesses 2
> Thank you for your valuable comments on 'Static vs. Dynamic Sparsity'. Here, we would like to make the following clarifications and supplementary explanations.
>
> Firstly, PureKV itself is not limited to static compression strategies. In fact, our framework has a high degree of flexibility and composability. As shown in the appendix, we combine PureKV with the dynamic KV compression method FastV: FastV retains all tokens in the first two layers and prunes some tokens in subsequent layers. After introducing PureKV, FastV performance has significantly improved. This fully demonstrates that PureKV can serve as a universal enhancement module, effectively improving various KV compression methods including static and dynamic methods.
>
> We compared it with the Top-p base method ZipVL. Although ZipVL achieves dynamic adjustment of token retention through the Top-p mechanism, it still needs to explicitly calculate attention scores at each layer to evaluate token importance and guide sparse attention patterns accordingly, without completely solving the problem of additional overhead caused by the need to calculate attention at each layer.

---

> ### Author Response · Authors · 2025-12-04
> **[2/3] Official Comment by Authors**
>
> # Weaknesses 3
> We have introduced more recent methods to comprehensively evaluate the performance advantages of PureKV, and have further expanded the comparison with the recent work in the revised manuscript.
>
> Specifically, the experiments in the paper have covered five representative baseline methods in the past two years, H2O (NeurIPS 2023), SnapKV (NeurIPS 2024), StreamingLLM (ICML 2024), FastV (ECCV 2024) and LOOK-M (EMNLP 2024 Findings). We have also added comparative experiments on AdaKV (NeurIPS 2025), PyramidKV (COLM 2025), and ZipVL (ICCV 2025). Although these methods propose novel ideas in dynamic KV cache compression, they still require explicit calculation of attention scores at each layer to evaluate the importance of tokens, making them incompatible with efficient attention mechanisms such as FlashAttention and sparse attention. ZipVL is compatible with sparse attention by performing an additional attention calculation, but it still cannot escape the computational overhead caused by calculating attention scores layer by layer. In contrast, PureKV achieves seamless integration with efficient attention mechanisms by cross layer importance estimating and only computing attention at the layer layer to guide high layer KV compression.
>
> In addition, we have added the evaluation results of the image understanding benchmark ChartQA and the long video understanding benchmark LongVideoBench, fully verifying the superiority of PureKV.
>
> Table 1.  Performance of KV cache compression strategy under a 20% KV cache budget based on Qwen2.5-VL-7B on MVBench. The best results are highlighted in **bold**. The second result is highlighted with an $\underline{underline}$.
> | Method| AC| AP| UA| OI| OS| AL| ST| SC| CO| CI| Avg.|
> |-|:-:|:-:|:-:|:-:|:-:|:-:|:-:|:-:|:-:|:-:|:-:|
> | Full Cache    | 0.7215 | 0.6995 | 0.5559 | 0.7014 | 0.1674 | 0.5052 | 0.7249 | 0.2859 | 0.3145 | 0.4595 | 0.5136 |
> | H2O| 0.6919 | 0.6812 | 0.4202 | 0.7025 | 0.1851 | 0.4587 | 0.6873 | 0.2672 | 0.3151 | 0.2852 | 0.4695 |
> | SnapKV| 0.6838 | 0.6933 | 0.4781 | 0.7153 | 0.1799 | 0.4595 | 0.7000 | 0.2924 | 0.2934 | 0.3124 | 0.4808 |
> | StreamingLLM| 0.6665 | $\underline{0.7005}$ | 0.5056 | 0.7563 | 0.1867 | 0.4675 | **0.7519** | 0.2046 | $\underline{0.3216}$ | 0.3802 | 0.4941 |
> | FastV| 0.7273 | 0.6973 | 0.5581 | 0.7127 | 0.1742 | 0.5111 | 0.7002 | 0.3095 | **0.3274** | $\underline{0.4366}$ | 0.5154 |
> | LOOK-M| 0.0074 | 0.0126 | 0.1268 | 0.0187 | 0.0014 | 0.0948 | 0.0292 | 0.0267 | 0.0000 | 0.2929 | 0.0610 |
> | PyramidKV| 0.7417 | 0.6978 | 0.3642 | 0.7054 | 0.1777 | $\underline{0.5546}$ | 0.6777 | $\underline{0.3302}$ | 0.2550 | 0.3624 | 0.4867 |
> | AdaKV | 0.7331 | 0.6974 | 0.5622 | $\underline{0.7260}$ | $\underline{0.1954}$ | 0.5515 | 0.6910 | 0.3223 | 0.2272 | 0.3528 | 0.5059 |
> | ZipVL  | **0.7538** | 0.7004 | $\underline{0.5654}$ | 0.7131 | 0.1821 | 0.5541 | 0.7035 | 0.3258 | 0.2475 | 0.4200 | $\underline{0.5166}$ |
> | PrueKV| $\underline{0.7490}$ | **0.7142** | **0.5624** | **0.7956** | **0.1957** | **0.5562** | $\underline{0.7242}$ | **0.3488** | 0.3086 | **0.4749** | **0.5429** |
>
> Table 2. Performance of KV cache compression strategy under a 20% KV cache budget based on Qwen2.5-Omni on AVSD. The best results are highlighted in **bold**. The second result is highlighted with an $\underline{underline}$.
> | Method| AVSD|
> |-|:-:|
> | Full Cache| 0.408587|
> | H2O| 0.416338|
> | SnapKV| 0.417255|
> | StreamingLLM  | $\underline{0.420158}$|
> | FastV| 0.414431|
> | PureKV| **0.426560**|
>
> Table 3. Performance of KV cache compression strategy under a 20% KV cache budget based on Qwen2.5-VL-7B on ChartQA. The best results are highlighted in **bold**. The second result is highlighted with an $\underline{underline}$.
> | Method| test_augmented | test_human1 |
> |-|:-:|:-:|
> | Full Cache    |     0.749133   |   0.436253  |
> | H2O|     $\underline{0.423131}$   |   $\underline{0.310473}$  |
> | SnapKV|0.274291   |   0.289163  |
> | StreamingLLM|     0.150806   |   0.212347  |
> | FastV|0.116688   |   0.175246  |
> | purekv|**0.451354**   |   **0.352862** |

---

> > ### Author Response · Authors · 2025-12-04
> > **[3/3] Official Comment by Authors**
> >
> > Table 4. Performance of KV cache compression strategy under a 20% KV cache budget based on Qwen2.5-VL-7B on LongVideoBench. The best results are highlighted in **bold**. The second result is highlighted with an $\underline{underline}$.
> > | Method| S2A    | O2E    | T2E    | T2O    | T2A    | O3O    | SSS    | SOS    | T3O    | TAA    | Avg.   |
> > |-|:-:|:-:|:-:|:-:|:-:|:-:|:-:|:-:|:-:|:-:|:-:|
> > | Full Cache    | 0.2062 | 0.3503 | 0.3863 | 0.2062 | 0.1085 | 0.3923 | 0.4494 | 0.4360 | 0.2915 | 0.3980 | 0.3225 |
> > | H2O| 0.1583 | 0.3407 | 0.3051 | 0.1292 | 0.0873 | 0.3191 | 0.3107 | 0.3950 | 0.2617 | 0.3450 | 0.2652 |
> > | SnapKV| 0.1608 | 0.3423 | 0.3153 | 0.1545 | 0.0744 | 0.3390 | 0.3768 | 0.4449 | 0.2395 | 0.3356 | 0.2783 |
> > | StreamingLLM  | 0.1811 | 0.3438 | 0.3603 | $\underline{0.1990}$ | 0.0947 | $\underline{0.3863}$ | 0.3419 | 0.5003 | 0.2738 | 0.3296 | 0.3011 |
> > | FastV    | $\underline{0.1860}$ | $\underline{0.3634}$ | $\underline{0.3644}$ | 0.1612 | $\underline{0.1102}$ | 0.3618 | $\underline{0.4350}$ | $\underline{0.4624}$ | $\underline{0.2810}$ | $\underline{0.3538}$ | $\underline{0.3079}$ |
> > | PureKV | **0.2182** | **0.3712** | **0.3842** | **0.2357** | **0.1107** | **0.3879** | **0.5572** | **0.4839** | **0.3240** | **0.3551** | **0.3428** |
> >
> > # Weaknesses 4
> > We first clarify the difference between PureKV and SeerAttention: SeerAttention belongs to sparse attention mechanism, which only accelerates the inference process and does not reduce the memory occupation of KV cache. The core goal of PureKV is to compress KV cache, thereby reducing memory consumption and accelerating inference.
> >
> > | Dimension               | SeerAttention                          | PureKV                                      |
> > |-------------------------|----------------------------------------|---------------------------------------------|
> > | **Type**                | Sparse Attention                       | KV Cache Compression                        |
> > | **Main objective**      | Inference acceleration                 | Memory optimization + inference acceleration|
> > | **Reduce memory?**      | ❌ (KV cache fully reserved)           | ✅ (5× compression)                         |
> > | **Train-free?**         | ❌ (Introduces learnable modules)      | ✅ (Completely train-free)                  |
> > | **Optimize granularity**| Block-wise                             | Token-wise, more flexible                   |
> > | **Compatibility**       | Output non-real attention and cannot be directly combined with KV compression methods such as H2O/SnapKV | Can be combined with and enhance H2O/SnapKV, and other KV compression methods |
> >
> > In addition, SeerAttention evaluates the importance of KV cache blocks at each layer through block-wise attention score calculation, while PureKV uses low layer attention scores to estimate the importance of high layers KV cache, directly discarding unimportant information to achieve more thorough acceleration and optimization.

---

### Official Review · Reviewer_vfdb · 2025-10-26

**Soundness:** 2
**Presentation:** 3
**Contribution:** 2
**Rating:** 4
**Confidence:** 4

**Summary:**

The authors propose PureKV, a plug-and-play framework for Vision-Language LLMs that jointly optimizes KV-cache compression and attention sparsity, remaining compatible with efficient kernels (e.g., FlashAttention/sparse attention). Core to the method is a cross-layer importance estimator that reuses shallow-layer attention scores and weights them by the L2 norm of deep-layer V vectors to select salient past tokens without computing deeplayer attention matrices. In addition, the authors designed a Spatial-Temporal Sparse Attention (ST-SpAttn) mechanism that purifies KV cache by suppressing spatial background noise and temporal redundancy. Experiments on VideoLLaMA2 and Qwen2.5-VL report up to 5x KV-cache compression and 3.16x prefill acceleration with small quality degradation, plus ablations and rank-correlation evidence supporting the cross-layer estimator. Overall, the work targets practical, system-aware inference gains for video VLLMs by coupling cache selection with sparsity design.

**Strengths:**

The paper is clearly written and easy to follow. It introduces a genuinely plug-and-play KV-cache framework that remains compatible with high-performance attention backends (e.g., FlashAttention). A lightweight cross-layer importance estimator—reusing shallow-layer attention and weighting deep-layer V by its L2 norm—efficiently ranks tokens, preserving speed while selecting the most salient cache entries.

**Weaknesses:**

The proposed method requires obtaining attention scores for KV-cache importance estimation, which appears incompatible with FlashAttention, as the latter does not explicitly compute or expose attention matrices.

**Questions:**

1. The explanation in Section 3.2 is somewhat unclear. The authors introduce Spatial-Temporal Sparse Attention (ST-SpAttn), claiming that it suppresses background noise and irrelevant visual distractions. However, the paper does not clearly explain why or how ST-SpAttn achieves this effect compared to full attention. Could the authors elaborate on the underlying mechanism and provide experimental evidence or visualizations to support this claim? Moreover, the section relies on qualitative descriptions without mathematical formulations or equations, which makes the method difficult to fully understand.

2. In Section 3.2, the authors state that existing KV-cache pruning strategies overlook structural modifications—specifically, that the KV cache at position j in layer i aggregates information from the first j + 1 tokens in layer i − 1. However, this issue seems straightforward to address by simply retaining the original position IDs. Could the authors clarify why this is considered a significant challenge and explain why preserving position information alone would not resolve the problem?

3. The experimental details are insufficient. For instance, the authors state that the proposed method reuses shallow-layer attention scores for deeper layers, but it remains unclear which specific shallow layers correspond to which deep layers. Clarifying this mapping or providing an illustrative example would help improve the reproducibility and understanding of the method.

4. The authors only present results on multimodal video understanding benchmarks, which is insufficient to demonstrate the generality of the proposed method. It would strengthen the paper to include additional experiments on image understanding benchmarks, such as VQAv2 [1] and ChartQA [2].

5. In Table 4, the results for the full-cache baseline are missing, making it difficult to assess the absolute performance and relative improvements of the proposed method.

Reference:

[1] Making the v in vqa matter: Elevating the role of image understanding in visual question answering. CVPR 2017.

[2] ChartQA: A Benchmark for Question Answering about Charts with Visual and Logical Reasoning. ACL 2022.

---

> ### Author Response · Authors · 2025-12-04
> **[1/2] Official Comment by Authors**
>
> # Weaknesses 1
> Good! Your idea coincides with this paper. The "KV cache importance estimation and FlashAttention compatibility" you pointed out is indeed a core challenge in the current field of efficient inference, and it is one of the challenges that this paper attempts to address.
>
> Modern attention accelerators such as FlashAttention achieve ultimate efficiency by integrating computation and memory access, avoiding explicit computation of attention matrices; However, traditional KV compression methods such as H2O and SnapKV rely on explicit attention scores layer by layer, which naturally conflicts with them.
>
> Our core contribution is to propose a KV Cache optimization framework that is compatible with efficient attention mechanisms: only calculating attention scores at the lower layer (such as layers 1-2), and using the lower layer attention scores to estimate the importance of high layers KV Cache, so that it can be compatible with efficient attention mechanisms such as FlashAttention when attention scores are not obtained at the high layers.
>
> # Questions 1
> The core idea of ST-SpAttn is to enforce the model to focus on the most critical Spatial-Temporal paths by means of structured sparsity, thereby avoiding information mixing in high layers.
> ### Mathematical Formalization: Accurate Description of ST-SpAttn
> Let the input video be represented as a token sequence $X \in \mathbb{R} ^ {T \times N \times d} $ , where:
> - $T $: Video frame rate
> - $N $: Number of tokens per frame
> - $d $: Hidden Dimension
>
> For the query vector $q_{t, i} $ at position $i $ of frame $t $, the total attention calculation is:
>
> $Attention_{full}(q_{t,i}) = \sum_{s=1}^{T} \sum_{j=1}^{N}Softmax \left( \frac{q_{t,i}^T k_{s,j}}{\sqrt{d}} \right)v_{s,j}$
>
> ST-SpAttn restricts the attention range by introducing a structured mask $M \in ${$ 0,1 $}$ ^{TN \times TN}$:
>
> $ST-SpAttn(q_{t,i}) = \sum_{(s,j) \in \mathcal{S}(t,i)} Softmax \left( \frac{q_{t,i}^T k_{s,j}}{\sqrt{d}} + m_{t,i,s,j} \right) v_{s,j}$
>
> Where $m_{t, i, s, j}=- \infty $ if $(s, j) \notin \mathcal{S}(t, i)$, otherwise it is 0.
>
> ##### The sparse set $\mathcal{S}(t, i) $ is defined as:
> 1. Spatial Sparse Path:
> $\mathcal{S}_{spatial}(t,i) =$ { $(1,j) \mid j = 1,\ldots,N $ } $ \cup $ { $(t,j) \mid j = 1,\ldots,N $ }
> - Contains all tokens from the first frame (global context anchor)
> - Contains all tokens of the current frame $t $(local spatial context)
>
> 2. Temporal Sparse Path:
> $\mathcal{S}_{temporal}(t,i) = $ {$(1,j) \mid j = 1,\ldots,N $} $ \cup $ {$ (t-1,i) $}
> - Include all tokens from the first frame
> - Token containing the same spatial position $i $from the previous frame
>
> Finally, $ \mathcal{S}(t,i)=$ $\mathcal{S}\_{spatial}$$(t,i)$ $ \cup  \mathcal{S}\_{temporal}(t,i) $ ,
>
> The information mixing problem of full attention: As shown in Figure 1 (c) of the paper, in standard causal self-attention, the KV cache at the jth position aggregates the information of the first j+1 tokens in the previous layer. This means that the information of unimportant tokens will be continuously mixed into the representation of important tokens through attention mechanisms. With the deepening of network layers, the entanglement between important and unimportant information will become increasingly severe, making it difficult for KV cache compression algorithms to accurately identify truly important tokens.
>
> # Questions 2
> Clarification on Position ID Retention: There is a viewpoint that retaining the original position IDs can maintain the structural integrity of the KV Cache. However, this only addresses the alignment issue of positional encoding and fails to resolve high layer semantic information mixing. Specifically, high layer KV vectors inherently aggregate multi-token information, leading to the mixing of important and unimportant data at high layers. This internal mixing cannot be corrected by external positional ID. ST-SpAttn mitigates this by constraining attention paths, reducing the aggregation of irrelevant information at the source, thereby producing purer and more compressible KV representations.
>
> # Questions 3
> In the paper, we use the attention score of the layer 2 to estimate the importance of subsequent layers of KV Cache starting from the layer 3.
> - We found through Spearman rank correlation coefficient analysis that the importance ranking of kvcache between different layers has a significant positive correlation. Figure 4 in the paper indicates that there is a significant positive correlation between the high layer importance ranking estimated at the lower layer (layer 1) and the true importance ranking at the higher layer, providing a solid theoretical basis for cross layer importance estimation.
> - We conducted a comprehensive hyperparameter analysis to investigate the effect of CLIE activation at different layers on PureKV performance. As shown in Figure 5, when the CLIE layer index is 2, PureKV performs the best.

---

> ### Author Response · Authors · 2025-12-04
> **[2/2] Official Comment by Authors**
>
> # Questions 4
> We have added the evaluation results of the image understanding benchmark ChartQA and the long video understanding benchmark LongVideoBench, and provided the results of the audio and video dataset AVSD in the appendix. In addition, we have also added the evaluation results of Qwen2.5-omni to demonstrate the generality of the method.
>
> Table 1. Performance of KV cache compression strategy under a 20% KV cache budget based on Qwen2.5-Omni on AVSD. The best results are highlighted in **bold**. The second result is highlighted with an $\underline{underline}$.
> | Method        | AVSD    |
> |---------------|:-------:|
> | Full Cache    | 0.408587|
> | H2O           | 0.416338|
> | SnapKV        | 0.417255|
> | StreamingLLM  | $\underline{0.420158}$|
> | FastV         | 0.414431|
> | PureKV      | **0.426560**|
>
> Table 2. Performance of KV cache compression strategy under a 20% KV cache budget based on Qwen2.5-VL-7B on ChartQA. The best results are highlighted in **bold**. The second result is highlighted with an $\underline{underline}$.
> | Method        | test_augmented | test_human1 |
> |---------------|:--------------:|:-----------:|
> | Full Cache    |     0.749133   |   0.436253  |
> | H2O           |     $\underline{0.423131}$   |   $\underline{0.310473}$  |
> | SnapKV        |     0.274291   |   0.289163  |
> | streamingllm  |     0.150806   |   0.212347  |
> | FastV         |     0.116688   |   0.175246  |
> | purekv        |     **0.451354**   |   **0.352862**  |
>
> Table 3. Performance of KV cache compression strategy under a 20% KV cache budget based on Qwen2.5-VL-7B on LongVideoBench. The best results are highlighted in **bold**. The second result is highlighted with an $\underline{underline}$.
> | Method        | S2A    | O2E    | T2E    | T2O    | T2A    | O3O    | SSS    | SOS    | T3O    | TAA    | Avg.   |
> |-|:------:|:------:|:------:|:------:|:------:|:------:|:------:|:------:|:------:|:------:|:------:|
> | Full Cache    | 0.2062 | 0.3503 | 0.3863 | 0.2062 | 0.1085 | 0.3923 | 0.4494 | 0.4360 | 0.2915 | 0.3980 | 0.3225 |
> | H2O           | 0.1583 | 0.3407 | 0.3051 | 0.1292 | 0.0873 | 0.3191 | 0.3107 | 0.3950 | 0.2617 | 0.3450 | 0.2652 |
> | SnapKV        | 0.1608 | 0.3423 | 0.3153 | 0.1545 | 0.0744 | 0.3390 | 0.3768 | 0.4449 | 0.2395 | 0.3356 | 0.2783 |
> | StreamingLLM  | 0.1811 | 0.3438 | 0.3603 | $\underline{0.1990}$ | 0.0947 | $\underline{0.3863}$ | 0.3419 | 0.5003 | 0.2738 | 0.3296 | 0.3011 |
> | FastV    | $\underline{0.1860}$ | $\underline{0.3634}$ | $\underline{0.3644}$ | 0.1612 | $\underline{0.1102}$ | 0.3618 | $\underline{0.4350}$ | $\underline{0.4624}$ | $\underline{0.2810}$ | $\underline{0.3538}$ | $\underline{0.3079}$ |
> | PureKV | **0.2182** | **0.3712** | **0.3842** | **0.2357** | **0.1107** | **0.3879** | **0.5572** | **0.4839** | **0.3240** | **0.3551** | **0.3428** |
>
> # Questions 5
> Thank you for pointing it out. We have added the results of the full-cache baseline to facilitate readers' evaluation of the proposed method.
>
> Table 4: Ablation study. CLIE: Cross-Layer Importance Estimation, ST-SpAttn: SpatialTemporal Sparse Attention, V: Weighted with L2 norm of V
> | CLIE | ST-SpAttn | V | Qwen2.5-VL | VideoLLaMA2 |
> |------|-----------|---|------------|-------------|
> | Full Cache |  |   | 0.7215   | 0.7676    |
> | ✗    | ✗         | ✓ | 0.7307     | 0.6985      |
> | ✓    | ✗         | ✓ | 0.7311     | 0.7020      |
> | ✓    | ✓         | ✗ | 0.7212     | 0.6936      |
> | ✓    | ✓         | ✓ | **0.7490** | **0.7588**  |

---

### Official Review · Reviewer_ieie · 2025-10-27

**Soundness:** 3
**Presentation:** 3
**Contribution:** 3
**Rating:** 6
**Confidence:** 3

**Summary:**

This paper presents a KV cache compression strategy and an attention module for video KV cache algorithms. The authors use several experiments to show the feasibility. Specifically, they reported that PureKV can achieve 5× compression and 3.16× prefill acceleration on VLLMs.

**Strengths:**

- The idea makes sense, combining two modules together for video KV cache compression with theoretically grounded cross-layer correlation analysis.
- The results are convincing, although more diverse video understanding tasks would strengthen the evaluation.
- The method achieves genuine compatibility with modern attention accelerators.

**Weaknesses:**

- The core techniques (attention-based importance scoring and sparse attention) are not novel individually; more critically, the selection of which layers to apply CLIE versus ST-SpAttn appears empirically driven rather than theoretically justified. A principled framework for determining optimal layer assignments would strengthen the contribution.
- While the authors briefly mention audio-visual experiments in the appendix (AVSD dataset), these results deserve fuller integration into the main evaluation. Testing on multimodal models like the Qwen2-Audio or Qwen2-Omni series would better demonstrate generalizability across modalities.
- The non-monotonic relationship between cache budget and performance (some tasks showing better results at 10% than 20% budget) warrants deeper investigation. Analyzing the distribution of retained tokens and their semantic properties could reveal why aggressive pruning sometimes outperforms moderate compression.
- The evaluation focuses on single-query scenarios, but real-world video understanding often involves multi-turn dialogue. Demonstrating how PureKV handles iterative questioning about the same video content and whether cache can be effectively reused across queries would significantly strengthen practical applicability claims.

**Questions:**

- Is there theoretical justification for the layer selection logic?
- Why do some tasks perform better at 10% cache budget than 20%? What tokens are being retained/dropped that cause this? A more detailed analysis may help.
- How does PureKV handle sequential questions about the same video? Can KV cache be reused across queries?
- Please consider adding more experiments, such as LongVideoBench and VideoMME, and indicate whether the method can be generalised well to the audio or audio-visual domain (for example, by testing on Qwen-3 Omni).

---

> ### Author Response · Authors · 2025-12-04
> **[1/2] Official Comment by Authors**
>
> # Weaknesses 1 and Questions 1
> Thank you for your attention to the novelty of our work. We would like to clarify that the **core contribution** of our work is not to propose isolated new modules, but to **build a KV cache optimization framework that is compatible with efficient attention mechanisms**, addressing the issue of existing KV compression methods relying on explicit calculation of layer by layer attention scores, which are incompatible with efficient attention mechanisms such as FlashAttention and SDPA. Both attention-based importance scoring and sparse attention are replaceable modules. We have combined PureKV with other KV cache methods in the appendix, significantly improving the performance of other methods and demonstrating the generalization ability and application value of this framework.
>
> The application layers of CLIE and ST-SpAttn are two configurable hyperparameters, and their selection is not purely empirical driven, but based on a deep understanding of Transformer architecture:
> - For CLIE: Thanks to the structural characteristics of residual connections, the high layer KV representation is essentially a refinement of lower layer semantics rather than reconstruction. We found through Spearman rank correlation coefficient analysis that the importance ranking of kvcache between different layers has a significant positive correlation. Figure 4 in the paper indicates that there is a significant positive correlation between the high layer importance ranking estimated at lower layer (such as the layer 1) and the true importance ranking at higher layer, providing a solid theoretical basis for cross layer importance estimation.
> - We conducted a comprehensive hyperparameter analysis to investigate the effects of CLIE and ST SpAttn activation at different layers on PureKV performance. As shown in Figure 5, when the CLIE layer index is 0, the PureKV performance significantly decreases. This indicates that layer 0 may not be able to capture sufficient contextual information, resulting in the inability to estimate cross layer importance. When the CLIE layer index is 2, PureKV performs the best, and VLLMs can use efficient attention mechanisms in higher layers to accelerate prefilling inference.
> - In addition, Figure 5 also reveals that ST-SpAttn typically leads to better performance when activated in high layers, but this advantage does not increase indefinitely with depth. The high layer KV cache mixes important and unimportant information, which is more suitable for spatiotemporal filtering. By applying ST-SpAttn in high layers, irrelevant visual interference and temporal redundancy can be effectively suppressed, ensuring that KV cache only retains more refined and structured information, ultimately improving the quality of compressed representations.
>
> # Weaknesses 2 and Questions 4
> Thank you for your curiosity and interest in the generalization ability of our method on other modalities. But it needs to be clarified that **the core task of PureKV** is to solve the KV cache explosion and inference delay problems faced by **Vision Language Large Models (VLLMs)** when processing visual modalities. Therefore, the experimental design in the main text focuses on typical visual understanding tasks and selects mainstream VLLMs such as VideoLLaMA2 and Qwen2.5-VL as evaluation models.
>
> Nevertheless, we also acknowledge the importance of multimodal generalization. Therefore, we have preliminarily explored the performance of PureKV in the Audio Visual Scene Dialogue (AVSD) task in the appendix, and further supplemented the experimental results on Qwen2.5-Omni. These additional experiments demonstrate that even for  Audio-Video LLMs, PureKV is still able to effectively identify and retain the multimodal KV cache that is most critical for generating responses, while maintaining answer quality through significant compression of the cache (such as 5x). This preliminarily validates the robustness and transferability of our method in a wider range of multimodal contexts.
>
> In addition, we have added the evaluation results of the image understanding benchmark ChartQA and the long video understanding benchmark LongVideoBench.

---

> ### Author Response · Authors · 2025-12-04
> **[2/2] Official Comment by Authors**
>
> Table 1. Performance of KV cache compression strategy under a 20% KV cache budget based on Qwen2.5-Omni on AVSD. The best results are highlighted in **bold**. The second result is highlighted with an $\underline{underline}$.
> | Method        | AVSD    |
> |---------------|:-------:|
> | Full Cache    | 0.408587|
> | H2O           | 0.416338|
> | SnapKV        | 0.417255|
> | StreamingLLM  | $\underline{0.420158}$|
> | FastV         | 0.414431|
> | PureKV      | **0.426560**|
>
> Table 2. Performance of KV cache compression strategy under a 20% KV cache budget based on Qwen2.5-VL-7B on ChartQA. The best results are highlighted in **bold**. The second result is highlighted with an $\underline{underline}$.
> | Method        | test_augmented | test_human1 |
> |---------------|:--------------:|:-----------:|
> | Full Cache    |     0.749133   |   0.436253  |
> | H2O           |     $\underline{0.423131}$   |   $\underline{0.310473}$  |
> | SnapKV        |     0.274291   |   0.289163  |
> | StreamingLLM|     0.150806   |   0.212347  |
> | FastV         |     0.116688   |   0.175246  |
> | purekv        |     **0.451354**   |   **0.352862**  |
>
> Table 3. Performance of KV cache compression strategy under a 20% KV cache budget based on Qwen2.5-VL-7B on LongVideoBench. The best results are highlighted in **bold**. The second result is highlighted with an $\underline{underline}$.
> | Method        | S2A    | O2E    | T2E    | T2O    | T2A    | O3O    | SSS    | SOS    | T3O    | TAA    | Avg.   |
> |-----|:------:|:------:|:------:|:------:|:------:|:------:|:------:|:------:|:------:|:------:|:------:|
> | Full Cache    | 0.2062 | 0.3503 | 0.3863 | 0.2062 | 0.1085 | 0.3923 | 0.4494 | 0.4360 | 0.2915 | 0.3980 | 0.3225 |
> | H2O           | 0.1583 | 0.3407 | 0.3051 | 0.1292 | 0.0873 | 0.3191 | 0.3107 | 0.3950 | 0.2617 | 0.3450 | 0.2652 |
> | SnapKV        | 0.1608 | 0.3423 | 0.3153 | 0.1545 | 0.0744 | 0.3390 | 0.3768 | 0.4449 | 0.2395 | 0.3356 | 0.2783 |
> | StreamingLLM  | 0.1811 | 0.3438 | 0.3603 | $\underline{0.1990}$ | 0.0947 | $\underline{0.3863}$ | 0.3419 | 0.5003 | 0.2738 | 0.3296 | 0.3011 |
> | FastV    | $\underline{0.1860}$ | $\underline{0.3634}$ | $\underline{0.3644}$ | 0.1612 | $\underline{0.1102}$ | 0.3618 | $\underline{0.4350}$ | $\underline{0.4624}$ | $\underline{0.2810}$ | $\underline{0.3538}$ | $\underline{0.3079}$ |
> | PureKV | **0.2182** | **0.3712** | **0.3842** | **0.2357** | **0.1107** | **0.3879** | **0.5572** | **0.4839** | **0.3240** | **0.3551** | **0.3428** |
>
> # Weaknesses 3 and Questions 2
> In fact, the non-monotonicity between cache compression rate and task performance was not first observed in PureKV, but has been repeatedly observed in multiple recent KV compression works, including H2O (NeurIPS 2023), LOOK-M (EMNLP 2024), VL Cache (ICLR2025), as well as Elastic Cache (ECCV 2024) and MadaKV (ACL 2025). H2O and VL Cache attribute this to the regularization effect introduced by KV cache pruning: moderate pruning can suppress redundant or noisy information, thereby enhancing the model's ability to focus on key information.
>
> # Weaknesses 4 and Questions 3
> Thank you for providing this highly practical suggestion. We fully agree that multi turn dialogue is a common and important use case in real video interaction scenarios.
>
> The current mainstream LLM inference efficiency research (including H2O, SnapKV, Streaming LLM, etc.) generally adopts single query as the default setting. Under this setting, PureKV significantly reduces memory usage and accelerates inference by identifying and retaining the KV cache that is most critical to the current problem at once, and evicting redundant parts. This has been fully validated on MVBench.
>
> For multi round dialogue scenarios, we further explain the adaptation mechanism of PureKV:
>
> Due to the fact that each round of questions may focus on different segments or semantic levels in the video (such as the first round asking "What action happened?" and the second round asking "What color of clothing did the protagonist wear?"), there are indeed differences in the required key visual contexts. For this purpose, PureKV can adopt the following strategies:
> - During each generation round, the most important KV cache is still dynamically selected based on the current query for inference, in order to maintain high-quality answers.
> - Do not immediately evict the KV cache judged as "unimportant" in this round, but retain the complete cache for reuse in subsequent rounds.
> - When the next round of queries arrives, re-evaluate the importance of KV cache and dynamically adjust the KV subset involved in the calculation.
>
> This mechanism ensures that:
> 1. Each round of inference can still enjoy the computational acceleration brought by sparse attention and KV compression.
> 2. To avoid performance degradation in subsequent rounds due to premature expulsion of potentially useful information.
> 3. No need to prefill the entire video from scratch for each query round, supports efficient cross round KV cache reuse.

---

### Official Review · Reviewer_n7gx · 2025-11-01

**Soundness:** 3
**Presentation:** 2
**Contribution:** 2
**Rating:** 4
**Confidence:** 5

**Summary:**

This paper introduces PureKV, a plug-and-play framework for the joint optimization of sparse attention and KV cache compression. PureKV features a lightweight token importance estimator that utilizes lower-layer attention scores and the L2 norm of value vectors to assess the importance of high-layer KV cache entries, ensuring compatibility with efficient attention mechanisms. Additionally, it presents a novel Spatial-Temporal Sparse Attention (ST-SpAttn) module tailored for video tasks, which purifies the KV cache by reducing spatial noise and temporal redundancy. Extensive experiments show that PureKV achieves up to 5× KV cache compression and 3.16× prefill acceleration, with negligible impact on quality.

**Strengths:**

1. The cross-layer importance estimation using lower-layer attention scores and value vector norms is both novel and empirically validated.
2. The introduction of ST-SpAttn for video tasks is well-motivated and demonstrates clear benefits in both cache purification and acceleration.
3. The experiments cover multiple VLLMs, tasks, and cache budgets, showing consistent improvements

**Weaknesses:**

1. Figure 1(c) could be further improved to more clearly illustrate the differences between dense attention and ST-SpAttn.
2. While the empirical results are strong, the theoretical justification for why lower-layer attention scores are sufficient for high-layer importance estimation could be further strengthened, for example by providing more formal analysis or theoretical bounds.
2. It would be beneficial to include an analysis of PureKV’s computational overhead in the ablation study.
3. H2O and StreamingLLM are not the strongest baselines; please consider including more recent methods [AdaKV](https://arxiv.org/pdf/2407.11550), [PyramidKV](https://arxiv.org/pdf/2406.02069), [HeadKV](https://arxiv.org/pdf/2410.19258?) and [PruneVID](https://arxiv.org/pdf/2412.16117) for a more comprehensive comparison.
4. The overall writing quality could be improved for greater clarity and readability.

**Questions:**

None

---

> ### Author Response · Authors · 2025-12-03
> **[1/2] Official Comment by Authors**
>
> # Weaknesses 1
> Thank you for your valuable suggestions on the clarity of the Figure. We have made the following improvements to Figure 1 (c) (d) in the revised version:
>
> We focus on the KV cache generation process of the last token to visualize its aggregated contextual sources under two attention mechanisms. In Dense Attention, the token indiscriminately focuses on all previous tokens in the sequence, causing information from the background, redundant frames, or irrelevant regions to constantly mix in at higher layer, resulting in highly entangled important and unimportant information, which is not conducive to subsequent KV cache compression strategies based on importance ratings.
>
> In contrast, in ST-SpAttn, the last token only focuses on the intra frame token and the token at the corresponding position in the previous frame. This sparse mode maintains contextual connections while effectively suppressing cross frame noise and spatial interference, thereby generating purer and clearer structured KV cache.
>
> # Weaknesses 2
> We explain from the perspectives of model architecture and statistical validation why lower layer attention scores are sufficient for high layer importance estimation:
> - Due to the structural characteristics of residual connections in Transformer architecture, high layer KV representation is essentially a refinement of low layer KV rather than reconstruction.
> - We found through Spearman's rank correlation coefficient analysis that the importance ranking of KV cache between different layers has a significant positive correlation. Figure 4 in the paper indicates that there is a significant positive correlation between the high layer importance ranking estimated at lower layer (such as the layer 1) and the true importance ranking at high layer, providing a solid theoretical basis for cross layer importance estimation.
>
> # Weaknesses 3
> We have added cost analysis for two core components of PureKV, CLIE (Cross Layer Importance Estimation) and ST-SpAttn, in our ablation study. As shown in table I, when CLIE is not enabled, attention scores need to be calculated at each layer, KV cache importance needs to be evaluated, and unimportant tokens need to be selected for compression. Although this reduces decoding latency, it increases prefilling latency. CLIE utilizes lower layer attention scores to estimate the importance of high layer KV cache, effectively alleviating prefilling latency. After further enabling ST-SpAttn's efficient attention, the prefill delay was reduced to 0.0355 ms/token, thanks to its structured sparse mode significantly reducing the amount of attention computation between visual tokens.
>
> Tabel I. Ablation study on inference speed under a 5% KV cache budget for PureKV on VideoLLaMA2.
>
> | Method          | CLIE | ST-SpAttn | Prefilling (ms/token) | Decoding (ms/token) |
> |-----------------|:----:|:---------:|:------------------:|:-----------------:|
> | Full cache      |  ✗   |     ✗     |       0.1190       |      36.73        |
> | PureKV       |  ✗   |     ✗     |       0.1523       |      28.45        |
> | PureKV      |  ✓   |     ✗     |       0.1241       |      28.21        |
> | PureKV     |  ✓   |     ✓     |       **0.0355**       |      **27.92**        |

---

> ### Author Response · Authors · 2025-12-03
> **[2/2] Official Comment by Authors**
>
> # Weaknesses 4
> We have introduced more recent methods to comprehensively evaluate the performance advantages of PureKV, and have further expanded the comparison with the recent work in the revised manuscript.
> Specifically, the experiments in the paper have covered five representative baseline methods in the past two years, H2O (NeurIPS 2023), SnapKV (NeurIPS 2024), StreamingLLM (ICML 2024) , FastV (ECCV 2024) and LOOK-M (EMNLP 2024 Findings). We have also added comparative experiments on AdaKV (NeurIPS 2025), Pyramid KV (COLM 2025), and ZipVL (ICCV 2025). Although these methods propose novel ideas in dynamic KV cache compression, they still require explicit calculation of attention scores at each layer to evaluate the importance of tokens, making them incompatible with efficient attention mechanisms such as FlashAttention and sparse attention. ZipVL is compatible with sparse attention by performing an additional attention calculation, but it still cannot escape the computational overhead caused by layer by layer attention calculation. In contrast, PureKV achieves seamless integration with efficient attention mechanisms by cross layer importance estimating and only computing attention at the layer layer to guide high layer KV compression.
> We explain the reasons for not comparing HeadKV and Prunevid as follows:
> - HeadKV is mainly aimed at long text tasks such as "Needle-in-a-Haystack". Its core idea is to infer the importance of each attention head based on whether the model can accurately recall specific keywords (i.e. "needles") in the input, and allocate cache budget accordingly. This method relies on literal matching between the input text and the output answer, making it difficult to directly transfer to VLLM settings for fair comparison.
> - PruneVID focuses on the compression of input visual tokens (i.e. token pruning), rather than KV cache compression. The goal is to reduce the number of visual tokens in the early stages of the model, thereby reducing the subsequent computational costs. This represents a different level of optimization compared to our work: token compression is applied to the input representation stage, while KV compression is applied to the autoregressive decoding stage. The two are not mutually exclusive but have the potential for synergy. In fact, how to effectively combine token compression, sparse attention, and KV cache compression is an important direction for our future work.
>
> Table II.  Performance of KV cache compression strategy under a 20% KV cache budget based on Qwen2.5-VL-7B on MVBench. The best results are highlighted in **bold**. The second result is highlighted with an $\underline{underline}$.
> | Method        | AC     | AP     | UA     | OI     | OS     | AL     | ST     | SC     | CO     | CI     | Avg.   |
> |---------------|:------:|:------:|:------:|:------:|:------:|:------:|:------:|:------:|:------:|:------:|:------:|
> | Full Cache    | 0.7215 | 0.6995 | 0.5559 | 0.7014 | 0.1674 | 0.5052 | 0.7249 | 0.2859 | 0.3145 | 0.4595 | 0.5136 |
> | H2O           | 0.6919 | 0.6812 | 0.4202 | 0.7025 | 0.1851 | 0.4587 | 0.6873 | 0.2672 | 0.3151 | 0.2852 | 0.4695 |
> | SnapKV        | 0.6838 | 0.6933 | 0.4781 | 0.7153 | 0.1799 | 0.4595 | 0.7000 | 0.2924 | 0.2934 | 0.3124 | 0.4808 |
> | StreamingLLM| 0.6665 | $\underline{0.7005}$ | 0.5056 | 0.7563 | 0.1867 | 0.4675 | **0.7519** | 0.2046 | $\underline{0.3216}$ | 0.3802 | 0.4941 |
> | FastV         | 0.7273 | 0.6973 | 0.5581 | 0.7127 | 0.1742 | 0.5111 | 0.7002 | 0.3095 | **0.3274** | $\underline{0.4366}$ | 0.5154 |
> | LOOK-M        | 0.0074 | 0.0126 | 0.1268 | 0.0187 | 0.0014 | 0.0948 | 0.0292 | 0.0267 | 0.0000 | 0.2929 | 0.0610 |
> | PyramidKV     | 0.7417 | 0.6978 | 0.3642 | 0.7054 | 0.1777 | $\underline{0.5546}$ | 0.6777 | $\underline{0.3302}$ | 0.2550 | 0.3624 | 0.4867 |
> | AdaKV | 0.7331 | 0.6974 | 0.5622 | $\underline{0.7260}$ | $\underline{0.1954}$ | 0.5515 | 0.6910 | 0.3223 | 0.2272 | 0.3528 | 0.5059 |
> | ZipVL  | **0.7538** | 0.7004 | $\underline{0.5654}$ | 0.7131 | 0.1821 | 0.5541 | 0.7035 | 0.3258 | 0.2475 | 0.4200 | $\underline{0.5166}$ |
> | PrueKV      | $\underline{0.7490}$ | **0.7142** | **0.5624** | **0.7956** | **0.1957** | **0.5562** | $\underline{0.7242}$ | **0.3488** | 0.3086 | **0.4749** | **0.5429** |
>
> # Weaknesses 5
> We will carefully summarize the comments of AC and all reviewers, make comprehensive revisions to the paper based on them.

---

### Author Response · Authors · 2025-12-04
**To AC and all reviewers**

We sincerely thank the reviewers for the positive assessment of our work，including its technical contributions of cross layer importance estimator and ST-SpAttn (n7gx, vfdb, xK7L)，novelty (n7gx)， clear writing (vfdb)，well-motivation (n7gx), Idea that make sense (ieie2),  convincing experiments results(n7gx, ieie2), clear benefits in both cache purification and acceleration (n7gx). We have carefully considered all comments and provide below our point-by-point responses and corresponding improvements to further strengthen the paper.

We regret that we were unable to discuss and clarify technical details with the reviewers during this review process. Please kindly ask AC to consider this objective limitation in the comprehensive evaluation.

Thanks to AC for hard work in this unexpected situation, and for reallocating and reviewing a large number of papers.

Thanks again to the reviewer for the careful reading and constructive feedback!

---

### Note · Authors · 2026-02-01

I have read and agree with the venue's withdrawal policy on behalf of myself and my co-authors.

---

### Meta-Review · Area_Chair_3R5r · 2026-01-07

**Summary:**

This paper proposes a method for KV cache optimization in vision language models. The paper argues KV cache optimization mechanisms are often incompatible with efficient attention modules such as FlashAttention and SparseAttention. Such modules achieve ultimate efficiency by integrating computation and memory access, avoiding explicit computation of attention matrices. However, traditional KV compression methods such as H2O and SnapKV rely on explicit attention scores layer by layer, which naturally conflicts with them.

The paper explores how to effectively identify and retain important KV cache entries while remaining fully compatible with efficient attention mechanisms. They aim to only calculate attention scores at the lower layer (such as layers 1-2), and use the lower layer attention scores to estimate the importance of high layers. Their method consists of two parts; PureKV computes attention scores only in the lower layer and leverages them to estimate the importance of KV cache in subsequent layers. Spatial-Temporal Sparse Attention (ST-SpAttn) with spatial and temporal sparsity specifically tailored for video understanding tasks.

The reviewers identified several strengths of the work, including clarity, novelty, lightweight and genuinely plug-and-play, effective decomposition of spatial and temporal redundancy, genuine compatibility with modern attention accelerators, and consistent improvements.

**Reviewer Concerns:**

Reviewer xK7L:
- **Adaptability of attention-based methods**: The reviewer requests discussion and comparison with methods such as NSA, MoBA, SeerAttention, and ZipVL. The authors argue that their focus is on training-free methods and NSA, MoBA, SeerAttention require training and ZipVL is compatible with sparse attention but does not eliminate the computational overhead caused by calculating attention scores layer by layer.
- **Static vs. dynamic sparsity**: The reviewer requests comparison with Top-p based methods and revision to the claim that all current methods are static. The authors clarify that PureKV is not limited to static compression strategies and provide comparison to the Top-p based method of ZipVL.  (to be added to the paper).
- **Lack of comprehensive comparisons**: The authors include new results comparing AdaKV (NeurIPS 2025), PyramidKV (COLM 2025), and ZipVL (ICCV 2025).  (to be added to the paper).
- **More comprehensive benchmarks**: The reviewer asks for evaluations on VideoMME, LongVideoBench, PerceptionTest, ActNet-QA, and NextQA. The authors provide evaluations on ChartQA and LongVideoBench (to be added to the paper).
- **Evaluation on prolonged video cases exceeding 20k context length**: The authors did not respond.
- **More extensive experiments on LLaVA-Video**: The authors did not respond.
- **Comparison to SeerAttention that adopt fine-grained token selection without discarding tokens**: The authors clarify by providing a comparison table.

Reviewer vfdb:
- **The proposed method requires obtaining attention scores**: The authors highlight that this is the limitation they try to tackle and they do so by only calculating attention scores at the lower layer such as layers 1-2.
- **Unclear description of ST-SpAttn in Section 3.2**: The authors provide a mathematical formulation (to be added to the paper).
- **Is it a major issues that KV-cache pruning strategies overlook structural modifications?** The authors provide an explanation (to be added to the paper).
- **Evaluation on VQAv2 and ChartQA**: The authors provide results on ChartQA and LongVideoBench.
- **Missing results in Table 4**: The authors provide results (to be added to the paper).

Reviewer ieie:
- **The core techniques are not novel**: The reviewer does not provide specific references for this and broadly says the method is empirically driven rather than theoretically justified. The authors note that their method can be combined with other KV cache methods and provide justification for their method based on the residual structure of Transformer architecture.
- **Audio-visual experiments are better in the main body than appendix**: The author said these are preliminary results and they wanted the focus on VLLMs while in the rebuttal they provide more audio-visual results with Qwen2.5-Omni.
- **The non-monotonic relationship between cache budget and performance warrants deeper investigation**: The authors argue this observation is not new and provide references.
- **Multi-turn video dialogue**: The authors argue that current literature generally adopts single query as the default setting but they also provide a description for how to adapt their method to multi-turn scenarios.
- **LongVideoBench and VideoMME**: The authors provide results on ChartQA and LongVideoBench.

Reviewer n7gx:
- **Figure 1(c) could be further improved**: The authors provide an explanation of how they have revised the Figure. The AC cannot confirm this as the PDF has not been updated.
- **Theoretical justification for why lower-layer attention scores are sufficient**: The authors provide an explanation based on the residual connections in Transformer architecture and the observation about the Spearman's rank correlation coefficient analysis.
- **Analysis of PureKV’s computational overhead**: The authors provide cost analysis for two components of their method.
- **Comparison to other baselines**: The reviewer asks for comparison to AdaKV, PyramidKV, HeadKV and PruneVID. The authors include results for H2O (NeurIPS 2023), SnapKV (NeurIPS 2024), StreamingLLM (ICML 2024) , FastV (ECCV 2024) and LOOK-M (EMNLP 2024 Findings) AdaKV (NeurIPS 2025), Pyramid KV (COLM 2025), and ZipVL (ICCV 2025).

**Reviewer Scores:**

Reviewers gave scores 4, 6, 4, 2. The reviewers shared concerns about limited comparisons to other baselines, and the description of the method including claims that do not fairly describe prior works or position the method accurately. None of the reviewers have responded during the rebuttal phase. The AC acknowledges the strengths of the work and the motivation for improving the compatibility of KV-cache optimization methods with efficient attention modules. The AC also acknowledges that the authors’ response has helped significantly towards resolving reviewers’ concerns. However, the AC still finds concerns that need to be resolved.

The writing and positioning of the paper needs to improve significantly. The response to the rebuttal requires a revision to the paper to clarify and correct problematic claims. However, the authors have not uploaded a revised draft during the rebuttal.

In terms of the description of the method, the AC finds the terminology unclear and inconsistent between the paper and the rebuttal. Is PureKV just the cross-layer importance estimation (CLIE) or both CLIE and ST-SpAttn? Figure 1(b) and all the writing up to the definition of the acronym CLIE on page 9 seem to suggest that PureKV is just CLIE but the rebuttal sometimes refers to PureKV as the combination of the two components (e.g. response to reviewer n7gx weakness 3). The paper needs to be revised to clarify the terminology.

The evaluation in the original paper was limited to MVBench with Qwen2.5-VL-7B. Reviewers requested evaluations on additional datasets and additional models including VQA evaluations and long video evaluations. The authors provided new results on ChartQA and LongVideoBench with Qwen2.5-VL-7B. They also provided results for Qwen2.5-Omni on AVSD but not for all methods. It is important to include a full suite of evaluations and models and include inference speed wall-clock time comparisons for each model and compare with inference speeds of related methods, particularly ZipVL. Moreover, even though part of the method is designed specifically for videos, the PureKV can still be evaluated on VQA tasks to provide a comprehensive view of the method. The authors may consider following ZipVL to include additional evaluations and models.

---

### Decision · Program_Chairs · 2026-01-26

Reject